# Internet of Things: A General Overview between Architectures, Protocols and Applications

**Marco Lombardi [1]**, **Francesco Pascale [2,\*]** and **Domenico Santaniello [1]**

1   Department of Industrial Engineering (DIIn), University of Salerno, 84084 Fisciano, Italy;
    malombardi@unisa.it (M.L.); dsantaniello@unisa.it (D.S.)
2   Department of Energy, Polytechnic of Milan, 20133 Milan, Italy
\*   Correspondence: francesco.pascale@polimi.it

**Abstract:** In recent years, the growing number of devices connected to the internet has increased significantly. These devices can interact with the external environment and with human beings through a wide range of sensors that, perceiving reality through the digitization of some parameters of interest, can provide an enormous amount of data. All this data is then shared on the network with other devices and with different applications and infrastructures. This dynamic and ever-changing world underlies the Internet of Things (IoT) paradigm. To date, countless applications based on IoT have been developed; think of Smart Cities, smart roads, and smart industries. This article analyzes the current architectures, technologies, protocols, and applications that characterize the paradigm.

**Keywords:** Internet of Things; machine to machine; smart vehicle; e-health; smart building; smart home; smart city; smart agriculture; Industry 4.0





## 1. Introduction

The Internet of Things (IoT) paradigm refers to a system of devices, interconnected with each other, equipped with computational capacity (smart objects), identifiable and enabled to transfer data over a network, without a required human interaction [1]. The concept behind this paradigm is the pervasive presence of smart devices, which by cooperating with each other and interacting with human beings achieve common goals [2].

Although this technology has started to be widely used only in recent years, it is possible to see traces of it already many years ago, even with theoretical hints. To give an example, in 1991, Mark Weiser wrote an article on ubiquitous computing: it is a model of human–machine interaction in which information processing is integrated within everyday objects rather than within individual personal computers [3]. One of the first real applications of a system similar to the described above can be found in the industrial sector, where realized machines were able to exchange information about their state independently. These systems were called machine to machine (M2M). The machines formed a closed system, and the primary purpose of the information exchange was to make the monitoring and management of the machines more efficient and less expensive. Compared to the current meaning of IoT, there was a lack of awareness of the potential that data could provide if reused in a broader context, for example, when aggregated with other systems connected through a network [4].

The term "Internet of Things" was first used in 1999 by Kevin Ashton during a Procter and Gamble presentation [5]. In this presentation, Ashton explained the possible benefits of using RFID technology in goods management. By equipping the goods with special devices, they could "communicate" information of interest (status, traceability, etc.). In this way, "things" and people could provide information about their status and the surrounding world, but in a much more efficient way.

The actual birth of the IoT dates back, according to Cisco estimates, to a period between 2008–2009, when for the first time, the number of connected objects exceeded

the world population. In 2010, the number of such objects had almost doubled compared to that time, reaching about 12.5 billion. Since those years, IoT, thanks to continuous technological developments and considerable investments by companies, has become increasingly widespread in everyday life.

According to IoT-analytics estimates, there are currently about 20 billion connected objects globally, and the IoT sector generates a market of about $150 billion. In 2024, connected objects will exceed 30 billion, and the market value will be about 1 billion. As with any new technology trend, there are three possible categories of challenges for IoT to overcome: business, society, and technology [6–8].

The business field's challenges mainly concern the identification of the motivation to start investing or not in a specific product and the design of a business model to achieve economic gain. In this category, depending on the use and the type of customer, products can be divided into three categories:

- Consumer IoT (smartphones, smart car, smartwatch, etc.);
- Commercial IoT (IoT Healthcare, Smart City, etc.);
- Industrial IoT (includes various types of devices for industrial use).

The challenges in society's field are to identify with the perspective of the customer who benefits from a product. To do this, it is necessary to consider some elements such as the constant change of requirements and demands imposed by the customer, the emergence of new devices, customer confidence in specific brands and products, and lack of knowledge of best practices in terms of privacy and security.

Although the current technologies that belong to the IoT domain can now be defined as advanced, several areas can be identified that need further development.

IoT needs minimal components to be integrated into everyday objects. The miniaturization and integration of components itself is a field that can expand with the integration of silicon components into metallic or fabric materials. In addition, there is a need for such components to quickly harvest the necessary energy from their surroundings and use it profitably. Smart objects need to withstand harsh conditions, be it humidity, temperature, or shock and vibration; for their everyday use, they also need to be extremely reliable, and guarantee very high and consistent quality. Another aspect that is often underestimated is the ability of smart devices to self-configure and organize themselves. Moreover, it will be necessary to find standard protocols to identify objects uniquely. Moreover, a critical field concerns security to find solutions to secure connected objects, preventing cyber-attacks that can undermine the global growth of the Internet of Things.

## 2. Most Common IoT Architectures

One of the main challenges to deal with the technological field to promote the deployment of IoT systems is to define a reference architecture that supports current features and future extensions. For this reason, such an architecture must be [9]:

- scalable, in order to manage the increasing number of devices and services without degrading their performance;
- interoperable, so that devices from different vendors can cooperate to achieve common goals;
- distributive, to allow to create of a distributed environment in which, after being collected from different sources, data are processed by different entities in a distributed way;
- able to operate with few resources, since objects generally have little computing power;
- secure so as not to allow unauthorized access.

Currently, there is no single reference architecture, and creating one is proving very complicated despite many standardization efforts. The main problem lies in the natural fragmentation of possible applications, each of which depends on many very often different variables and design specifications. This problem must be added to each supplier's

tendency to propose its platform for similar applications [9–11]. In Figure 1, it is possible to see some of most common IoT Architecture used.

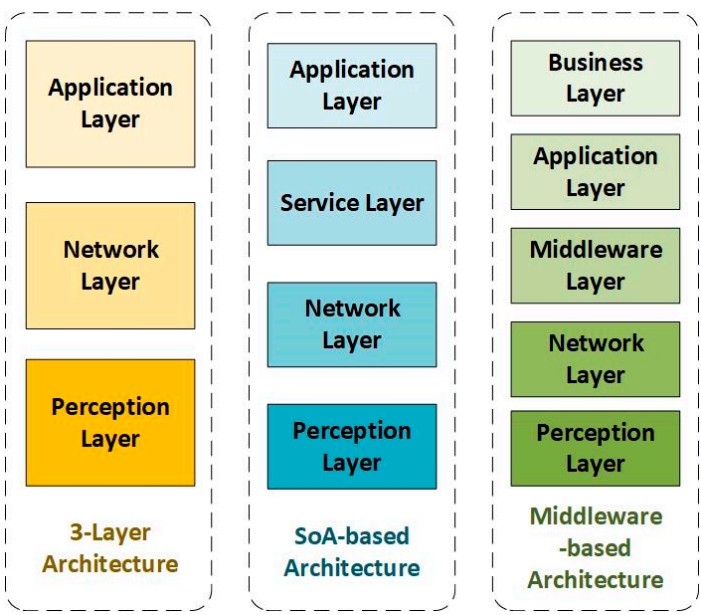

**Figure 1.** Most common IoT architectures.

*2.1. Three-Level Architecture*

A generic high-level architecture composed of three layers has been introduced in the literature [10]:

- Perception, which represents the physical layer of objects and groups all the features;
- Network, which represents the communication layer responsible for the transmission of data to the application layer through various technologies and protocols;
- Application, which represents the application layer in which the software offering a specific service is actually implemented.

2.1.1. Perception Layer

The perception layer represents the physical level of objects and interacts with the surrounding environment by collecting and processing information. This level includes objects that, being able to interact with the external world and being equipped with computing capabilities, become in a certain sense "intelligent" or "smart", where smart refers to the technological aspects (the smart technologies used), while intelligent refers to the functional aspects (self-identification, self-diagnosis, self-testing, etc.) of the sensor [9,10].

These smart objects, which are the fundamental blocks on which the IoT is based, can be objects of common usage (a refrigerator, a television, a car, etc.) or simple devices equipped with sensors and computing capabilities. In general, smart objects are equipped with the following essential properties [12,13]:

- Communication: objects can connect to each other and to resources on the Internet to use data and services, update their status, and cooperate to achieve common goals;
- Identification: objects must be uniquely identified.

Depending on the specific application, one or more of the following properties may also be added:

- Addressability: objects can be directly reachable, i.e., addressed, to be interrogated and/or configured remotely;
- Sensing and actuation: objects can collect information about the surrounding world and manipulate it through the use of sensors and actuators;

- Embedded information processing: the smart objects are equipped with calculation capabilities to process the results of the sensors and drive the actuators;
- Localization: objects are aware of their physical location or can be located;
- User interface: objects can communicate appropriately with users via displays or other interfaces.

Table 1 lists some of the technologies used to implement smart objects' various features [11]. There are many hardware platforms on the market equipped with these features, among the many examples are RaspberryPi, Arduino, Beaglebone Black, etc.

**Table 1.** Most used technologies in IoT field.

| Scheme 4 | Used Technologies |
|---|---|
| Communication | Zigbee, Bluetooth, Wifi, Near Field Communication (NFC), Radio-Frequency IDentification (RFID), etc. |
| Identification | Electronic Product Code (EPC), Ubiquitous Code (uCode), Quick Response (QR), etc. |
| Addressability | IPv4, IPv6 |
| Sensing e Actuation | Micro Electro-Mechanical Systems (MEMS) e Micro-Opto-Electro-Mechanical Systems (MOEMS), embedded sensors, etc. |
| Embedded information processing | Field Programmable Gate Array (FPGA), Programmable Logic Controller (PLC), microcontrollers, Single-board computer, System-on-Chip (SoC) |
| Localization | Global Position System (GPS), Galileo, etc. |
| User interface | Displays, remote control, etc. |

### 2.1.2. Network Layer

The network layer has the task of transporting the data provided by the perception level to the application layer. It includes all the technologies and protocols that make this connection possible and should not be confused with the network layer of the ISO/OSI model, which only routes data within the network along the best path [9].

There are a large number of protocols that can be used in IoT. Table 2 shows some of the most used protocols, grouped according to the ISO/OSI model. Each of them has pros and cons, and the use must be evaluated according to the application. For example, the IPv6 protocol was born first to solve the problem of addressing space (which with the old IPv4 protocol was about to run out) and then to ensure scalability for the systems. However, this protocol is designed for wired networks. To meet wireless sensor networks (Wireless Sensor Network or WSN), the 6LoWPAN protocol [11] was created. WSN is, in fact, composed of devices characterized by low computational power that often have to minimize energy consumption. In this way (with the 6LoWPAN protocol), the devices can directly overlook the network without needing other devices that act as gateways. In specific applications, however, such gateways may be necessary. In these cases, it is possible to use data link layer protocols, such as Bluetooth or ZigBee, to connect gateways and sensors, and then only interface the gateway to the network using the IPv6 protocol network layer.

In this layer, wireless protocols are particularly important. Compared to those requiring cables, wireless sensors can be installed in hard-to-reach environments and require less material and human resources for installation. Additionally, in a wireless sensor network, the various nodes can be added or removed easily, and their location can be changed without reconsidering the structure of the entire network. The choice of a protocol to use depends on the network's size, the power consumption of each node, and the transmission speed needed in a given application.

In other applications, however, it may be necessary to build a wired network. The latter enjoys more excellent reliability and higher transmission rates [14]. To give an example, it is possible to think of a vehicle's internal network that connects the various Electronic Control Units (ECU) that control the mechanical parts of the car (steering, brake, etc.). In this case, it is essential to have a reliable and fast network, because delays or malfunctions could have severe consequences for the people on board the car.

**Table 2.** Main protocols used in the IoT field.

| Application Layer | | | |
|---|---|---|---|
| CoAP, MQTT, AMQP, XMPP, DSS | | | |
| Service Discovery: mDMS, DNS-SD, SSDP | | | |
| Security: TLS, DTLS | | | |
| **Transport Layer** | | | |
| TCP, UDP | | | |
| **Network Layer** | | | |
| Addressing: IPv4/IPv6 | | Routing: RPL, CORPL, CARP, etc. | |
| **Adaption Layer** | | | |
| 6LoWPAN, 6TiSCH, 6Lo, etc. | | | |
| **Data Link Layer** | | | |
| IEEE 802.15.4 (ZigBee, etc.) | IEEE 802.15.1 (Bluetooth) | LPWAN (LoRaWAN, etc.) | RFID, NFC |
| IEEE 802.11 (WiFi) | IEEE 802.3 (Ethernet) | IEEE 1901 (PLC) | Z-Wave |
| **Physical Layer** | | | |

### 2.1.3. Application Layer

The application layer includes all the software necessary to offer a specific service. In this level, the data from the previous levels are stored, aggregated, filtered, and processed, and databases, analysis software, etc., are used. As a result of this processing process, the data is made available to real IoT applications (smart wearable, smart car, etc.). This is often done using certain software defined as middleware, which has the task of hiding the heterogeneity of the underlying layers. Some software technologies currently widely used to manage the vast amount of data provided by devices are:

- Cloud computing, where services such as storage or data processing are provided from a set of pre-existing resources, configurable and remotely available in the form of distributed architecture;
- Edge computing, where data processing is partially distributed on the peripheral nodes of the network to increase the performance of IoT systems.

The management of the format of the data to be processed also belongs to this level. These can be of type [11]:

- binary-based, small size but not readable by human beings;
- text-based, larger size but readable by human beings.

Among the many commercial platforms used for implementing IoT applications, some examples are Amazon AWS, Microsoft Azure, Xively, Firefox WebThings Gateway, etc.

### 2.2. Service-Oriented Based Architecture

The last architecture presented is service-oriented based architecture. In general, service-oriented architecture (SoA) is a component-model based, which can be designed to connect different functional units of applications though interfaces and protocols [15,16]. SoA is designed to coordinate services and enable the reuse of software and hardware components. SoA can be easily integrated into IoT architecture, extending the three-layer architecture, adding a new layer between network layer and application layer, called thes service layer, which provide services to support the application layer. This represents the four-layer SoA-Based IoT architecture, in which there are the perception layer, the network layer, the service layer, and finally the application layer. Here there is a new layer that integrates the SoA functionalities [17,18], and it is different from previously described architecture. The service layer consists of service discovery, service composition, service management, and service interfaces. Service discovery is used to discover the service requests; service composition is used to interact with the connected objects and integrate services to get the requests in an efficient way; service management is used to manage and

determine the trust mechanisms to understand the service requests; service interfaces are used to support interactions among all provided services.

### 2.3. Middleware Architecture

Another important and very common architecture in IoT is middleware based IoT architecture or five-layer architecture [19]. In the recent years, the proposed architecture for IoT needs to address many factors like scalability, interoperability, reliability, QoS, etc. In this regard, middleware based IoT architectures assist in creating applications more efficiently; this layer acts as a connection between applications, data, and users. In fact, IoT development depends on the technology progress and design of various new applications and business models [20]. To allow these characteristics and many others, a five-layer architecture has been proposed, which is composed of five levels: perception layer, network layer, middleware layer, application layer, and business layer. In particular, the middleware layer has some critical functionalities, such as aggregating and filtering the received data from the hardware devices, performing information discovery, and providing access control to the devices for applications.

In general, middleware is a software or service programming that can provide an abstraction interposed between IoT technologies and applications. In middleware the details of different technologies are hidden, and the standard interfaces are provided to enable developers to focus on the development of applications without considering the compatibility between applications and infrastructures. The middleware is gaining more and more importance in the last years due to its major role in simplifying the development of new services and the integration of legacy technologies into new ones. This excepts the programmer from the exact knowledge of the variegate set of technologies adopted by the lower layers. Advantages are:

1. Support on various applications;
2. Runs on various operating systems and platforms;
3. Distributed computing and the interaction of services among heterogeneous networks, devices, and applications;
4. Support standard protocols;
5. Provides standard interfaces, providing portability and standard protocols to enable interoperability, and making middleware play an important role in standardization;
6. Provides a stable high-level interface for applications.

### 2.4. Other Used Architectures

The high-level architecture just described provides a simple and general overview of an IoT project's structure and has been described in more detail to provide an overview of the leading technologies and protocols used in IoT. Depending on the application, however, it could be necessary to add additional layers or adapt the architecture to the specific application to realize.

In particular, there are two different architectures: the architectures belonging to the first class are obtained by adapting to the context of IoT existing architectures; the architectures belonging to the second class are built from scratch. Some examples of architectures can be seen in Table 3.

**Table 3.** Example of IoT architectures.

| Architectures Adapted to the IoT Context | Architectures from Scratch |
| --- | --- |
| IETF Protocol stack | Cloud-based architectures |
| Server-based architecture | Edge computing-based architectures |
| | Social Internet of Things (SIoT) architectures |

### 2.4.1. Internet Engineering Task Force (IETF) Protocol Stack

In this architecture, as can be seen in Figure 2, the authors try to adapt the suite of TCP/IP protocols to the context of IoT [9].

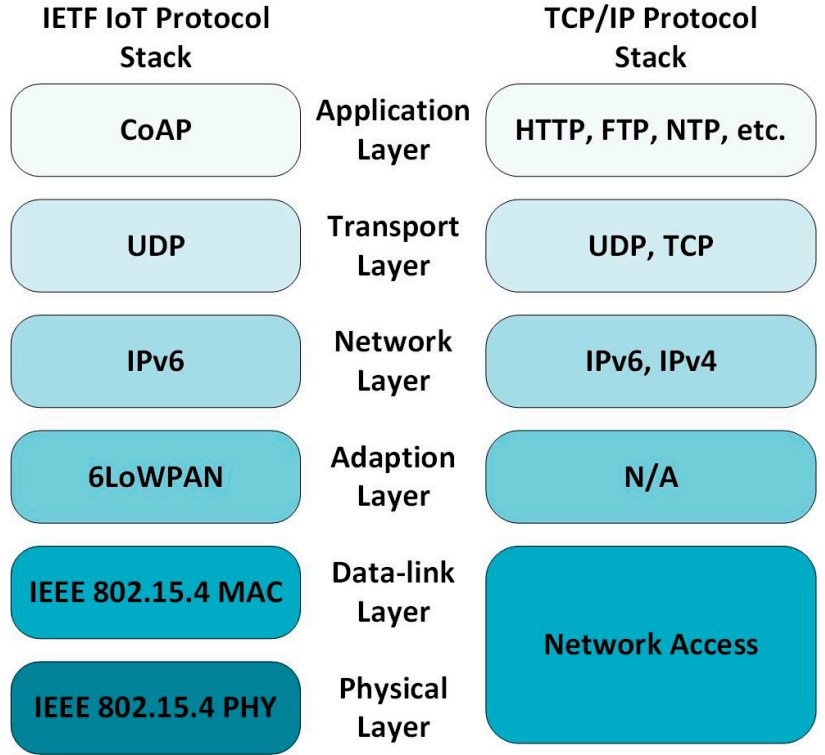

**Figure 2.** IETF protocol stack for IoT (**left**) and TCP/IP stack (**right**).

Due to the scarcity of computational resources and the heterogeneity of devices and traffic, new protocols have been introduced that have replaced or flanked the TCP/IP stack protocols.

In particular, instead of the application-level protocols of the TCP/IP stack, the Constrained Application Protocol (CoAP) is used, which represents a lightweight version of the Hypertext Transfer Protocol (HTTP), suitable to working with devices and sensors with limited resources. CoAP also uses the User Datagram Protocol (UDP) at the transport level, which, compared to the Transmission Control Protocol (TCP) used by HTTP, offers fewer services but is much lighter [21]. Finally, a layer is added (adaptation layer) in which the IPv6 packet headers, using the IPv6 over Low Power Wireless Personal Area Network (6LoWPAN) protocol, are encapsulated and compressed so that devices can manage them with little computing power. These protocols will be discussed in detail in the next section.

### 2.4.2. Server-Based Architecture

In this type of architecture, the various devices are connected to a gateway server that manages the devices' connection to the Internet, shown in Figure 3.

The main difference from the architecture described above is that the devices are not directly accessible and only look at the Internet through the gateway. In this way, the constraints given by the scarcity of computational resources and heterogeneity must be managed only at the lowest level, where the connection between the devices and the gateway itself is implemented. At the higher level, the standard protocols of the TCP/IP stack can be used.

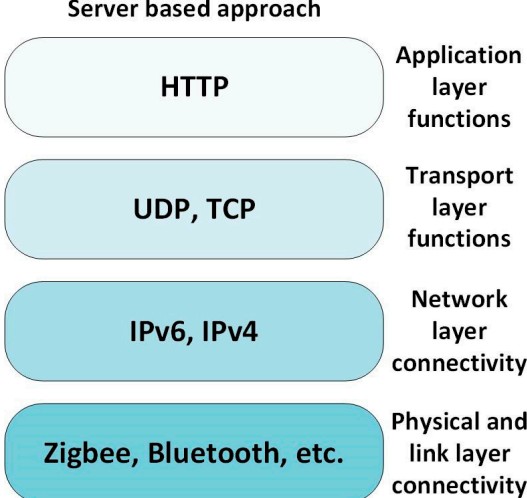

**Figure 3.** Server-based architecture.

### 2.4.3. Cloud-Based Architectures

In this type of architecture, the large amount of data generated by objects is stored, processed, and presented to the user through services made available from the cloud [22] There are many architectures of this type realized by various companies such as AWS by Amazon, Azure by Microsoft, Xively by Google, etc. In this paper, the Amazon AWS architecture will be illustrated [23].

AWS consists of four main elements: The Device Gateway, the Registry, the Rules Engine, and the Device Shadow. The Device Gateway acts as an intermediary between the connected devices and the cloud services through the Message Queue Telemetry Transport (MTTQ) protocol. MTTQ is a lightweight, publish–subscribe messaging protocol designed for situations where low power consumption is required, and the available bandwidth is limited. Security between devices is ensured by using the Transport Layer Security (TLS) protocol. The Rules Engine processes incoming messages and distributes them to other devices or cloud services. The Registry is responsible for assigning a unique ID to each connected device, regardless of device type, vendor, or connection mode. Additionally, it is responsible for storing metadata about connected devices in order to track them. Device Shadows are virtual representations of physical objects. These representations are persistent and stored in the cloud to be accessed at any time (even when the device is offline) by cloud services or other devices.

### 2.4.4. Edge Computing-Based Architectures

Nowadays, the concept of IoT is often placed side by side with Edge Computing. However, Edge Computing represents the future of IoT, because it has been proposed to be integrated with IoT to enable computing services devices, aiming to increase the performance and resilience of services [24]. Edge Computing is a horizontal architecture that provides distributed computing, data storage, and control capabilities. It interferes between cloud structures and end users to make available files and resources that are usually accessible only through a network connection. Thanks to the advantage of distributed architecture, Edge Computing can provide faster response and greater quality of service [25]. Building an architecture based on Edge Computing may have more benefits:

- Minimizes latency; many actions are taken very close to where the action is;
- Allows for bandwidth saving, avoiding over dimensioning the band to the Cloud;
- Solves some security issues, because many decisions are taken in a subnet and are not exposed to the risks arising from the external Internet.

The Edge Computing applications are multiple and differentiated, and it is possible to monitor or analyze in real time network data of industrial devices. It can be possible, for

example, to have actions machine to machine (M2M), perform actions on other machines or Human Machine Interaction (HMI), launch alerts, and report to specific users. In Figure 4, there is an example of fog/edge computing architecture:

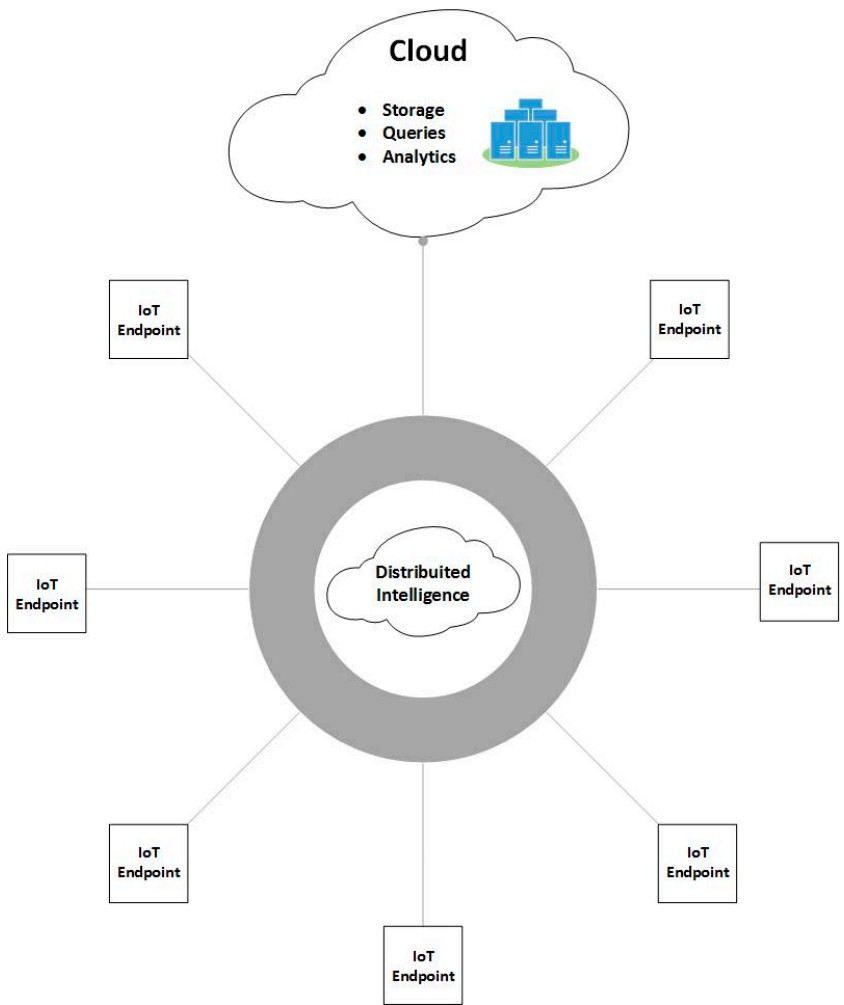

**Figure 4.** Edge Computing-based architecture.

The problems related to data transmission delays and security in data management have led to the development of Edge Computing as a potential solution. Edge Computing, defined as IT architecture in which computing applications, data-storage, and services are completely or partially pushed near the end-user [26], could bring a significant improvement in IoT paradigm.

Edge Computing architectures for IoT (ECAs-IoT) include security-based architectures, which also include Software Defined Networking (SDN) based approaches. Machine learning based architectures include Transferring Trained Models (TTM) and Hierarchical Fog-Assisted Computing Architecture (HiCH). There are architectures oriented to data placement strategy; a remarkable example are those dedicated to Fog infrastructures, architectures that are oriented to Big Data processing. Finally, there are orchestration-based architectures that contain SDN-based architectures and architectures based on Cloudlet and Fog Computing.

Today, challenges facing Edge Computing in IoT are made possible by the advent of new communication technologies and hardware, such as 5G, and the components of modern IoT devices, which are becoming smaller and performing.

2.4.5. Social Internet of Things Architecture

In a social network based IoT architectures (or SIoT), objects are listed on a social platform creating groups and relationships (as in a traditional social network) in order to provide certain services [27]. Such an approach has some advantages:

- The structure of SIoT can be modeled to ensure the navigability of the network. The discovery of objects and services can be performed effectively, and the scalability is also guaranteed, as in the case of social networks formed by people;
- A level of reliability can be defined to establish the degree of interaction with objects;
- The models developed for the study of typical social networks made up of people can be reused.

In this architecture, it is possible to see three main elements: the SIoT server, the gateway, and the objects. Each of the elements can be composed of three layers: sensing (or perception), network, and application. These have similar functionalities to those described in the high-level architecture. The server is responsible for managing and storing data (base sub-layer), providing the main functionalities of the siot system (component sub-layer), and offering interfaces for objects and people (interface sub-layer).

In general, the gateway only transfers data between different networks, but depending on the objects it has to manage, it may have a set of features covering all three layers. On the other hand, the objects have only the task of collecting data from the surrounding domain, but, as in smartphones, for example, they could also perform higher-level functions.

## 3. The Enabling Technologies and Most Common Protocols

The IoT concept in the real world can be realized through the integration of several enabling technologies. In this section, the most relevant enabling technologies for IoT are presented, focusing on the different layers listed above. At the bottom of all architecture, we can find the perception layer, which contains all physical devices and objects: in this layer, the main function is to identify and track objects. To achieve this function, several technologies can be implemented: one on the most common ones for this layer is RFID, used to identify and track objects without contact. It supports data exchange via radio signals over a short distance [28,29], and it works through RFID tag, RFID reader, and antenna [30]. Tags are characterized by a unique identifier and are applied to objects; readers trigger the tag transmission by generating an appropriate signal, which represents a query for the possible presence of tags in the surrounding area and for the reception of their IDs. Each RFID tag is attached in an object and has its unique identification number. An RFID reader can identify an object and obtain the corresponding information by querying the attached RFID tag through appropriate signals. RFID can be useful in the perception layer of IoT to identify and track objects and exchange information. Another important category of technologies is the wireless sensor network, which can monitor and track the status of devices and transmit the status data to the control center or sink nodes via multiple hops [31,32]. Sensor networks consist of a certain number (which can be very high) of sensing nodes communicating in a wireless multi-hop fashion; the collected data are analyzed to take specific actions based on required services. The IoT sensors can be smart sensors, actuators, or wearable sensing devices. Other technologies are: Single Board Computers (SBCs) integrated with sensors and built-in TCP/IP and security functionalities, which are typically used to realize IoT products (e.g., Arduino Yun, Raspberry PI, Beagle Bone Black, etc.); and RFID sensor network (RSN), which has the possibility of supporting sensing, computing, and communication capabilities in a passive system.

In the middle of IoT architecture we can find the network layer, which is used to define routing and provide data transmission support through integrated heterogeneous networks. Examples of communication protocols used for the IoT are Wi-Fi, Bluetooth, IEEE 802.15.4, Z-wave, and LTE-Advanced. Some specific communication technologies also used are RFID, Near Field Communication (NFC), and ultra-wide bandwidth (UWB). RFID is the first technology used to realize the M2M concept (RFID tag and reader): the RFID reader transmits a query signal to the tag and receives reflected signal from it,

which in turn is passed to the database, and it identifies objects based on the reflected signals [33]. RFID tags can be active, passive, or semi-passive/active, and they have a range from 10 cm to 200 m. The NFC works at high frequency band at 13.56 MHz and supports data rate up to 424 kbps; the applicable range is up to 10 cm [34]. The UWB communication technology is designed to support communications within a low range coverage area using low energy and high bandwidth, whose applications to connect sensors have been increased recently [35]. Another communication technology is Wi-Fi that uses radio waves to exchange data among things within 100 m range [36]. Bluetooth presents a communication technology that is used to exchange data between devices over short distances using short-wavelength radio to minimize power consumption [37]. Bluetooth special interest group (SIG) produced Bluetooth 4.1, which provides Bluetooth Low Energy as well as high-speed and IP connectivity to support IoT. LTE (Long-Term Evolution) is originally a standard wireless communication, and it can cover fast-travelling devices and provide multicasting and broadcasting services. LTE-A (LTE Advanced) [38] is an improved version of LTE including bandwidth extension, which supports up to 100 MHz, downlink and uplink spatial multiplexing, extended coverage, higher throughput, and lower latencies.

In many architectures, it can be possible to find a service layer such as SoA-based architecture. The service layer is located between the network layer and the application layer and provides efficient and secure services to objects or applications. In the service layer, the following enabling technologies should be included to ensure that the service can be provided efficiently: interface technology, service management technology, middleware technology, and resource management and sharing technology. The interface technology must be designed in the service layer to ensure the efficient and secure information exchange for communications among devices and applications. To support applications in IoT, an interface profile (IFP) can be considered as a service standard, which can be used to facilitate the interactions among services provided by various devices or applications. To achieve an efficient IFP, universal plug and play should be implemented [39,40]. Service management can effectively discover the devices and applications, and schedule efficient and reliable services to meet requests. A service can be considered as an integrated application, including collection, exchanging, and storage of data, or an association of these behaviors to achieve a special objective [41].

Various heterogeneous networks are integrated to provide data delivery for all applications in IoT (smart transportation, smart grid, etc.). To reduce the cost, some applications can share part of the network resources to increase their utilization. In this case, ensuring that information requested by various applications is delivered on time is a challenging issue in IoT. Existing resource sharing mechanisms primarily focus on the spectrum sharing, which is used to efficiently coordinate multiple networks in the same frequency to maximize the utilization of network resources [42,43].

Applications are on the top of the architecture, exporting all the system's functionalities to the final user. However, this layer is not considered to be part of the middleware, but exploits all the functionalities of the middleware layer. Using standard web service protocols and service composition technologies, applications can realize a perfect integration between distributed systems and applications. Many IoT standards are proposed to facilitate and simplify application programmers' and service providers' jobs. Different groups have been created to provide protocols in support of the IoT. In this paragraph, some of the most common protocols that can enable a reliable and secure communication in IoT are presented as shown in Table 4 [44,45].

**Table 4.** Most common protocols in IoT.

| Protocol | Layer |
|---|---|
| IEEE 802.15.4 | Perception Layer |
| LoWPANs | Network Layer |
| ZigBee | Network Layer |
| Z-wave | Network Layer |
| LoraWLAN | Network Layer |
| Sigfox | Network Layer |
| NB-IoT | Network Layer |
| Message Queue Telemetry Transport (MQTT) | Application Layer |
| Constrained application protocol (CoAP) | Application Layer |
| Extensible messaging and presence protocol (XMPP) | Application Layer |
| Data distribution service (DDS) | Application Layer |
| Advanced message queuing protocol (AMQP) | Application Layer |

- IEEE 802.15.4 is a protocol designed for the physical layer and the MAC layer in wireless personal area networks (WPANs). This protocol is used to focus on low-rate WPANs, providing low rate connections of all things in a personal area with low energy consumption, low rate transmission, and low cost [46].
- Low-power WPANs (LoWPANs) are organized by many low-cost devices connected via wireless communications (Tan and Koo 2014). In comparison with other types of networks, LoWPAN has several advantages (small packet sizes, low power, low bandwidth, etc.). 6LoWPAN protocol (an enhancement of LoWPANs), designed combining IPv6 and LoWPAN, has several advantages: great connectivity and compatibility with legacy architectures, low-energy consumption, ad-hoc self-organization, etc.
- ZigBee is a wireless network technology, designed for short-term communication with low-energy consumption and great reliability. In ZigBee protocol, five layers are included: physical layer, MAC layer, transmission layer, network layer, and application layer.
- The main objective of Z-wave is to provide reliable transmission between a control unit and one or more end-devices; Z-wave is suitable for networks with low bandwidth. Although both ZigBee and Z-wave support the shortrange wireless communication with low cost and low energy consumption, there are some differences between them.
- LoRaWAN is a cloud-based MAC (Media Access Control) layer protocol but primarily serves as a network layer protocol for managing communications between LPWAN (Low Power Wide Area Network) gateways and end-node devices such as routing protocol, managed by LoRa Alliance. Version 1.0 of the LoRaWAN specification was released in June 2015. LoRaWAN defines the communication protocol and system architecture for the network, while the physical LoRa layer allows long-range communication link. LoRaWAN is also responsible for managing communication frequencies, data rates, and power for all devices. Devices in the network are asynchronous and transmit when they have data available for sending. Data transmitted by a device (called an endpoint) are received by multiple gateways, which forward data packets to a centralized network server (or network server). The network server filters out duplicate packets, performs security checks, and manages the network. Data are then forwarded to the application servers. The technology shows high reliability for moderate load; however, it does present some performance issues related to sending acknowledgments.
- The Sigfox standard is based on ultra narrow band RF communication with very low consumption. It takes advantage of the 868 MHz frequency and is not subject to concessions. The possible applications are countless: for example, the remote detection of sensors and meters.
- Narrowband Internet of Things is an LPWAN radio technology standard developed by 3GPP to enable communication for a wide range of cellular devices and services,

the specifications of which were frozen in 3GPP Release 13, in June 2016. Other 3GPP IoT technologies include eMTC and EC-GSM-IoT.

- Message Queue Telemetry Transport (MQTT) uses a publish/subscribe technique: it is a messaging protocol, which is used to collect measured data on remote sensors and transmit them to servers. MQTT is a simple and lightweight protocol and supports networks with low bandwidth and high latency.
- Constrained Application Protocol (CoAP) is a messaging protocol based on representational state transfer (REST) architecture [47,48]. CoAP has been proposed to modify some HTTP functions to meet the requirements for IoT; in fact, it is an application layer protocol in the 6LoWPAN protocol stack and aims to enable resources constrained devices to achieve RESTful interactions.
- Extensible Messaging and Presence Protocol (XMPP) is an instant messaging protocol based on XML streaming protocols. XMPP inherits features from XML protocol, so it has great scalability, addressing, and security capabilities, and it can be used for multiparty chatting, voice and video streaming, and tele-presence.
- Data distribution service (DDS) is a publish/subscribe protocol for supporting high performance device-to-device communication. DDS has been developed by object-manage-group and is a data centric protocol, in which multicasting can be supported to achieve great QoS and high reliability.
- Advanced Message Queuing Protocol (AMQP) is an open standard message queuing protocol used to provide message service (queuing, routing, security, reliability, etc.) in the application layer; it focuses on message-oriented environments and can be considered as a message-oriented middleware protocol.

## 4. Applications in IoT

As said previously, there are many applications field of the IoT like Big Data, Cloud Computing, health care, Smart City, Smart Home, Smart Grid, mobile application, cyber industries, Smart Agriculture, automotives, and many others.

The importance of IoT has gone established with time, especially for those applications natively interconnected such as mobile application, and it lends itself to all those pervasive/ubiquitous computing applications [49]. In fact, there is often confusion between the concept of IoT and Context-Aware Computing, because they seem closely linked. Although these two concepts are often used simultaneously, they remain two separate things. IoT is one of the enabling technologies of context-aware computing, and today, a large number of IoT-based pervasive systems take advantage of Context-Awareness as a core feature [50–52].

Another important paradigm that fits with the IoT is Industry 4.0, which represents the current trend of automation and data exchange in manufacturing technologies. Industry 4.0 includes cyber–physical systems, the Internet of things, and Cloud Computing. Today, several research works explain this concept. Indeed, Guo-jian Cheng et al. talk about Industry 4.0 Development and its application for intelligent manufacturing [53]. This concept is strict in relationship with IoT, because to realize the vision of cyber–physical systems, it iss necessary that all systems are interconnected over the network, and IoT provides the right technology base. As highlighted by the EU Framework Program for Research and Innovation (HORIZON 2020) (Horizon 2020 is the financial instrument implementing the Innovation Union, a Europe 2020 flagship initiative aimed at securing Europe's global competitiveness.), many funds will be invested by the European community in IoT and Industry 4.0, because this will soon represent a sure source of work for companies [54].

The concept of IoT is often associated with Smart City and Smart Home. In fact, Smart City has defined as an urban development vision that integrate multiple information and communication technologies to improve the quality of life [55]. The assets that are included and managed in this vision are multiple; for example, we can consider local departments information systems, schools, libraries, transportation systems, hospitals, power plants, water supply networks, waste management, and law enforcement. Smart

Home is defined as a modern home that has appliances, lighting, and/or electronic devices that can be controlled remotely by the owner, often through mobile. As evidence, these two concepts integrate with the IoT paradigm and are enabled by it. K. E. Skouby and P. Lynggaard suggest a four-layer model that joins and interfaces these elements by deploying technologies such as 5G, Internet of Things, Cloud of Things, and distributed artificial intelligence [56]. Talari et al. provide an inclusive review on the concept of the smart city besides their different applications, benefits, and advantages. In this work, it can be possible find a review of IoT technologies and their capabilities to merge into and apply to the different parts of smart cities [57].

5G represents a new era technology that will allow us to surf at speeds much higher than the current ones; in fact, if 4G technology provided an average of 10 Mbps per user, 5G will make us navigate at a minimum of 100 Mbps [58]. This will certainly give more possibilities for mobile devices and a greater impetus to the development of new planning strategies such as smart cities and all that follows, such as having more and more objects connected to each other and able to exchange information. Obviously, this great range of additional opportunities is undoubtedly a great point in favor of this technology.

Another important and challenging application for IoT is the Smart Grid system. The Smart Grid aims to use energy power in a safe and correct way in which the power supply system can distribute electricity avoiding waste. For these reasons, today the use of Smart Grid and Smart Micro Grid is very widespread. An example of this application is presented by Clarizia et al., who proposed an architecture of a real-time energy management system that provides several Smart Meters [59]. This application monitors connected loads and communicate with a Smart Concentrator via CAN Bus that stores the data forwarded from a single Smart Meter, in order to make this information available for http request/response. Li et al. present the design and implementation of a novel co-simulator and evaluate the effectively IoT-aided algorithms for scheduling the jobs of electrical appliances [60].

During last year, health care support systems are growing up, and these are essential for the medical support, with an IoT communication framework as the main enabler for distributed worldwide health care applications. Vitali and Pernici have proposed an approach to a health-care scenario enriched with IoT devices; these techniques allow the discovery of interconnections between processes and external factors which have an impact on them [61]. Ahouandjinou et al. have explored the opportunities for IoT to realize the vision of the future of health care to attainment a new monitoring status system for patients of the Intensive Care Unit (ICU), to improve medical care service performances [62].

Today, to support IoT, it is important to consider all the substructures that are used to support the IoT devices. In particular, Cloud Computing is one of technologies that is more applied and better lends itself to be used in IoT. It is important for example for data storing and sharing between various IoT devices, such as communication from and to sensor and actuator or mobile device. Galache et al. presents the Cloud project, whose main aim is making citizens aware of city resources and helping them to use and care these resources by means of smart IoT services in the Cloud [63]. Carrillo et al. present a framework used to provide the needed computational power to the Smart Building by using Cloud Computing to have all computational power as well as control monitor capabilities in the cloud [64].

As its definition, Big Data represents large and complex data sets, which include analysis, capture, data curation, search, sharing, storage, transfer, visualization, querying, updating, and information privacy. In last few years, the IoT challenge consists in managing and analyzing the entire large amount of data generated by sensors all around the world. This is the reason why the concept of Big Data is often associated with that of IoT. Sezer et al. propose a combined framework that brings Big Data, IoT, and semantic web together to build an augmented framework. They provide a realistic use case that demonstrates how the model can implement the desired functionality and achieve the goals of such a model [65]. Bashir et al. present an IoT Big Data Analytics IBDA framework for the

storage and analysis of real time data generated from IoT sensors deployed inside the smart building, developed by using Python and the Big Data Cloudera platform [66].

Another important concept associated with IoT is Smart Agriculture. Starting from concept of precision agriculture, that is an integrated system of methodologies and technologies designed to increase crop production, quality, and productivity of a farm, we have the definition of Smart Agriculture. It is based on the aim "do the right thing, in the right place at the right time, with the right amount", respecting the real needs of plants. Indeed, it defines the scope of application solutions aimed at the monitoring, management, and optimization of different processes related to agriculture. Baranwal et al. propose an integrated approach based on IoT device, capable of analyzing the sensed information and then transmitting it to the user [67]. This device can be controlled and monitored from remote location, and it can be implemented in agricultural fields, grain stores, and cold stores for security purpose. This study is oriented to accentuate the methods to solve problems like identification of rodents, threats to crops, and delivering real time notification based on information analysis and processing without human intervention. Kapoor et al. describe an approach to combine loT and image processing to determine the environmental factor or man-made factor (pesticides/fertilizers), which is specifically hindering the growth of the plant [68].

No less important than the previous ones are the Blockchains. Blockchains represent the reference infrastructure for the operation of this network of "intelligent objects". Many think that distributed trust technology is the only technology capable of ensuring scalability, respect for privacy, and reliability for growing IoT environments [69,70].

Blockchains are a candidate for the role of key application for the IoT. The technology in question can be used to track billions of connected devices, allowing the processing of the transactions they produce and the coordination between physical devices. This decentralized approach would eliminate the failure points of traditional networks, facilitating the creation of a more resilient ecosystem in which smart devices can operate. Furthermore, the cryptographic algorithms used by Blockchains would allow to increase the protection of private consumer data [71].

One of the most interesting applications is the Cognitive IoT based on Blockchain as shown in [72]. Cognitive IoT is the use of cognitive computing technologies in combination with data generated by connected devices and the actions those devices can perform [73,74]. In this way, in a computer, system understanding means being able to take in large volumes of both structured and unstructured data and derive meaning from them—that is, establish a model of concepts, entities, and relationships. This provides a wealth of major and interesting applications.

The sensors' rapid spread and their produced data also affected the industry of automotive. In fact, for a few years it is possible to see a fast increase of technologies on board of a car both for control and assistance systems than for monitoring and diagnostic systems. In this scenario, IoT offers an important support for automotives, since it furnishes the best way to help the automobile manufacturers thanks to its paradigm implementations. It is possible to affirm that the Automotive Internet of Things (IoT) is an emerging research field that applies the IoT to the intelligent transport system [75]. With the introduction of smartphones, cloud, Edge Computing, and mobile Internet, the automotive ecosystem is shifting toward the Internet of Vehicles (IoV) [76,77]. These technologies have brought the car to become a very smart vehicle, so now the automobile is considered an integrated and complicate ecosystem of objects that cooperates with each other.

Trying to summarize, the applications of most significant interest then turn out to be:

- Smart Wearable: are wearable devices, low energy consumption, equipped with sensors and networked to collect data on users in order to monitor daily physical activity (to keep track of progress) or other information useful for health (heartbeat, quantity, and quality of sleep, etc.) to identify possible problems on time and do prevention;
- Smart Vehicle and Connected Vehicle: systems that were once mechanical (steering, brakes, etc.) have been replaced by electronic control units (ECUs) able to commu-

nicate with each other and with the outside world. This is to efficiently exploit the fuel and increase the driver's safety by monitoring parameters of interest thanks to electronics. The internet connection can be used to monitor traffic (and then choose the most appropriate route to get to a specific destination or find parking quickly), send signals in case of failure, receive timely assistance, and have access to infotainment services and many other applications. In this case, the ECUs form a network of smart objects inside the car, but also the car itself, seen in a broader context (like a smart city), becomes a smart object;

- e-Health: the goal is to monitor patients' health through devices (which can also be placed inside the human body) to make prevention and make diagnoses and treatments even when patients are far from the hospital. Additionally, monitoring the demand for healthcare services makes it possible to invest efficiently in specific areas of healthcare;

- Smart Building and Smart Home: smart objects can be used to monitor the structural integrity of buildings (and thus ensure more excellent safety), environmental parameters (such as temperature or humidity), or the presence of people in specific places. All this to make the environment comfortable and at the same time efficiently manage light, electricity, heating, etc.;

- Smart City: a network of sensors can be used to efficiently manage water resources, transport, energy, waste collection, etc., which would reduce pollution and waste and increase the comfort of citizens. Two possible examples could be intelligent parking management and public lighting management. In the first case, citizens could avoid wasting time looking for a free parking space, and the pollution caused by the car would be reduced; in the second case, the lighting could be managed according to the transit of pedestrians and vehicles, which would allow energy savings;

- Smart Metering and Smart Grid: thanks to the monitoring of the electricity grid, it is possible to manage the distribution and generation of energy (even that generated by small generators such as wind or photovoltaic plants scattered throughout the territory);

- Smart Agriculture (or precision agriculture): thanks to a network of sensors and actuators, you can monitor the health and the actual needs of crops to exploit resources (water, fertilizers, etc.) in an efficient and targeted way;

- Smart Factory (or Industry 4.0): by integrating new technologies in production processes, working conditions could be improved (an example could be the support of a robot to the human operator) as well as safety and productivity in an industry.

## 5. Technological Challenges in the IoT Field

However, there are other technological challenges on which research is focusing to promote the development and diffusion of IoT [78,79]:

- Standardization: in order to ensure the development of the IoT, it is crucial to have open standards for the connectivity of systems, interoperability of the various elements, etc. This process would facilitate both technological innovation (thanks to the public availability of standards) and independence from specific technologies or vendors;

- Availability and reliability: data must be available anytime and anywhere for each authorized object. Therefore, mechanisms are needed for the object's interoperability and the transfer and restoration (in case of unexpected data events). Additionally, the mobility of devices must be managed appropriately;

- Data storage, processing, and visualization: New methods must be found to efficiently manage and visualize the vast amount of data coming from smart objects;

- Scalability: research on this topic serves to make it possible to add new services and devices to existing IoT systems without degrading their presentations. In particular, it is necessary to take into account constraints such as memory, computing power, bandwidth, etc.;

- Management and self-configuration: the user can efficiently manage a large number of devices. Additionally, smart objects must be able to self-configure in response to external events as much as possible, always in order to simplify management;
- Unique identification of smart objects: each smart object must have unique identification in order to be reached by all the others. This process represents a problem, especially concerning the considerable increase of smart objects present;
- Energy consumption: smart objects must manage energy efficiently. They often communicate with other devices via wireless technologies and have batteries as a power source that cannot always be replaced easily;
- Security and privacy: as far as security is concerned, in general, we have that: Communications between devices are often wireless, and this makes it easy to "eavesdrop"; the low computing capacity they have makes it challenging to implement elaborate countermeasures. As far as privacy is concerned, one of the main problems lies in the fact that smart objects collect a considerable amount of information about users; a possible attacker could have access not only to users' data but also to their habits and information about their health, etc. Moreover, in an IoT application, the concepts of "safety" (security of physical objects and people) and "security" (security of data and information systems) tend to converge on the same level, since the objects have gained the ability to interact with the surrounding world.

## 6. Conclusions

This article has provided a current overview of architectures, technologies, protocols, and applications that characterize the Internet of Things (IoT) paradigm. In particular, the main architectures used in the IoT domain have been described based on their reference layers. The leading enabling technologies and most common protocols are IEEE 802.15.4, LoWPANs, ZigBee, Z-wave, and LoraWLAN, and Sigfox No-IoT, MQTT, CoAP, XMPP, DDS, and AMQP have been discussed. The main application areas have been identified, such as Big Data, Cloud Computing, health care, Smart City, Smart Home, Smart Grid, mobile application, cyber industries, Smart Agriculture, and automotive.

The global IoT market is growing exponentially. The statistics show that it reached 745 billion dollars in 2019. However, to realize its growth, it is unnecessary to analyze data, statistics, and market forecasts. In fact, it is before our eyes how important connected devices have become and how emerging technologies such as the Industrial Internet of Things, smart homes, smart cities, smart factories, and smart metering are already a reality.

The possibilities for the development of the Internet of Things are endless. Much will depend on the programmatic lines that global market manufacturers will adopt to make the connected devices as compatible as possible and thus increase the degree of interoperability and integration without neglecting the aspects related to safety. The analysis companies' forecasts are optimistic: in 2023, the IoT market globally should reach 318 billion dollars.

**Author Contributions:** Conceptualization: M.L., F.P. and D.S.; methodology: M.L., F.P. and D.S.; formal analysis: M.L.; investigation: F.P.; resources: D.S.; data curation: F.P.; writing—original draft preparation: M.L. and D.S.; writing—review and editing: F.P.; visualization: M.L. and D.S.; supervision: F.P. All authors have read and agreed to the published version of the manuscript.

**Funding:** This research received no external funding.

**Institutional Review Board Statement:** Not applicable.

**Informed Consent Statement:** Informed consent was obtained from all subjects involved in the study.

**Data Availability Statement:** Not applicable.

**Conflicts of Interest:** The authors declare no conflict of interest.

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
