# Peer review of "Internet of Things: A General Overview between Architectures, Protocols and Applications"

_information, doi:10.3390/info12020087_

Round 1
Reviewer 1 Report
The authors presented a well-written manuscript, described their manuscript in a good way, however they missed some important applications of IoT to discuss such as
1- the role of incorporating the blockchain for IoT.
2-Cognitive IoT and its applications
3- Cognitive radio based IoT
Authors are encouraged to use the following references to improve their manuscript adding some paragraphs about the above-mentioned applications.
1- Novo, Oscar. "Blockchain meets IoT: An architecture for scalable access management in IoT." IEEE Internet of Things Journal 5.2 (2018): 1184-1195.
2-Wu, Mingli, et al. "A comprehensive survey of blockchain: From theory to IoT applications and beyond." IEEE Internet of Things Journal 6.5 (2019): 8114-8154.
3-Lao, Laphou, et al. "A survey of IoT applications in blockchain systems: Architecture, consensus, and traffic modeling." ACM Computing Surveys (CSUR) 53.1 (2020): 1-32.
4-Awin, Faroq A., et al. "Technical issues on cognitive radio-based Internet of Things systems: A survey." IEEE Access 7 (2019): 97887-97908.
5- Park, Jin-ho, et al. "CIoT-Net: a scalable cognitive IoT based smart city network architecture." Human-centric Computing and Information Sciences 9.1 (2019): 29.
6- Saghiri, Ali Mohammad, et al. "A framework for cognitive Internet of Things based on blockchain." 2018 4th International Conference on Web Research (ICWR). IEEE, 2018.
Author Response
Thanks for the valuable comments and suggestions aiming at improving the overall quality of this paper.
We have made the following changes:
1) In section 4 we have added a brief overview on Blockchain in IoT and Cognitive IoT and their possible applications

Reviewer 2 Report
Will be better to include Sigfox, NB-IoT, 5G and describe them with more details.
Author Response
Thanks for the valuable comments and suggestions aiming at improving the overall quality of this paper.
We have made the following changes:
1) In section 3 we have added a description of Sigfox and NB-IoT. In section 4 we have added a brief overview on 5G and its potentialities.

Reviewer 3 Report
A good historical retrospective of the development of IoT is given in the article at its beginning.
Since the article is an overview, some improvements can be made.
Business and society challenges are briefly depicted (lines 57-70), however nothing is said about technological challenge as one of the “three possible categories of challenges for IoT” (line 55).
Cited reference number 8 (on line 56) is about “inductive output tube – IOT” and it does not concern to the IoT - Internet of Things.
It is good to mention a difference between intelligent and smart sensors (line 111): “An intelligent sensor is able to detect conditions and respond to them. Intelligent sensors should not be confused with smart sensors, where "smart" refers to being technologically advanced. A smart sensor may have advanced features, but it does not have conditional functions which make it "intelligent."[ https://www.techopedia.com/]”
The title of Table 1 needs to be changed: "many" is not appropriate here.
According to the fig. 1 and the logic of the paper the sections 2.2 and 2.3 must be reconfigured, this means that 2.2 has to be „SOA-based…“ and 2.3 – „Middleware“. In the discussion of the Middleware architecture should mention the advantages of this approach, as well as the need for its development.
All figures are disproportionate to the text. On the fig. 2 the 6LoWPAN belongs to AdAptation layer, however on the table 2 it is in AdOptation. Correction is needed.
The section “2.4.4 Edge computing-based Architectures” is not mentioned on the Table 3, maybe it is good to extend the table.
Some repetitions in the text of the paper article can be avoided by better structuring the article. For example, the Section “3 The enabling technologies and most common protocols” mixes many topics - technologies and protocols, they need to be separated. Paragraphs are too long. I would suggest following Table 1 to be more consistent and clearer, or following the architectural layers as perception, network and service layers are discussed here. So, it is better to divide the whole section into subsections.
At first glance, calling the barcode identification technique as IoT technology (line 399) along with the most common protocols seems strange. It is better to refine the description of the identification techniques.
The text from 445 to 465 should be moved to the explanation of the middleware architecture and added to the current section 2.2.
For reasons of readability standards and protocols such as IEEE 802.15.4, LoWPAN, Zigbee, etc., it is better after listing them, as done from 482 to 534, to correlate them with the appropriate levels according to Table 4, which is not mentioned anywhere in the text.
Section 4 is not balanced, a more specific description of the applications would be useful, not just their registering. “Applications in IoT” is a very wide topic. The article is an overview, with a very extensive bibliography, so this section can be expanded with more specifics.
It would be useful to have a summary on IoT development in recent years and some exchange of views from numerous reviews from the past. There is no conclusion and section dedicated to the discussion of what has been written.
Author Response
Thanks for the valuable comments and suggestions aiming at improving the overall quality of this paper.
We have made the following changes:
1) In the introduction section, we have included a paragraph concerning the technological challenges of IoT.
2) We have corrected citation number 8, which was added in error.
3)We have better clarified the difference between Smart Sensor and Intelligent Sensor.
4) We have fixed all the typing errors, style errors, and paragraph layout reported
5) We have better specified the advantages and needs regarding the use of Middleware-based architecture.
6) All the suggested revisions on figures and tables have been done.
7) We have modified section 3, as indicated, trying to make it easier to understand.
8)To avoid confusion, we have eliminated the paragraphs related to barcode as an identification technique.
9)We have made some changes by expanding section 4

Round 2
Reviewer 3 Report
Most of my remarks are considered in the revised manuscript.
There is still no Conclusions section, as required by the journal.
Author Response
Thanks for the valuable comments and suggestions aiming at improving the overall quality of this paper.
We have made the following changes:
1) Added chapter 6 conclusions